# Indirect Measurement of β-Glucan Content in Barley Grain with Near-Infrared Reflectance Spectroscopy

**DOI:** 10.3390/foods11131846

**Published:** 2022-06-23

**Authors:** Roberta Ghizzoni, Caterina Morcia, Valeria Terzi, Alberto Gianinetti, Marina Baronchelli

**Affiliations:** Council for Agricultural Research and Economics (CREA), Research Centre for Genomics and Bioinformatics (GB), Via S. Protaso 302, 29017 Fiorenzuola d’Arda, PC, Italy; roberta.ghizzoni@crea.gov.it (R.G.); caterina.morcia@crea.gov.it (C.M.); valeria.terzi@crea.gov.it (V.T.); marina.baronchelli@crea.gov.it (M.B.)

**Keywords:** β-glucans, barley, NIRS, health-related molecules, soluble dietary fiber

## Abstract

β-Glucan is a component of barley grains with functional properties that make it useful for human consumption. Cultivars with high grain β-glucan are required for industrial processing. Breeding for barley genotypes with higher β-glucan content requires a high-throughput method to assess β-glucan quickly and cheaply. Wet-chemistry laboratory procedures are low-throughput and expensive, but indirect measurement methods such as near-infrared reflectance spectroscopy (NIRS) match the breeding requirements (once the NIR spectrometer is available). A predictive model for the indirect measurement of β-glucan content in ground barley grains with NIRS was therefore developed using 248 samples with a wide range of β-glucan contents (3.4%–17.6%). To develop such calibration, 198 unique samples were used for training and 50 for validation. The predictive model had R^2^ = 0.990, bias = 0.013% and RMSEP = 0.327% for validation. NIRS was confirmed to be a very useful technique for indirect measurement of β-glucan content and evaluation of high-β-glucan barleys.

## 1. Introduction

Barley grains contain a noticeable amount of β-glucan (generally, from about 2% to 10% [1]), a kind of mostly soluble fiber that makes them suitable to produce foods with healthy functions [2]. The β-glucan, chemically a population of (1→3)-(1→4) mixed linked β-glucans, is a non-starchy polysaccharide representing the major structural component of the barley endosperm cell walls [3]. Specifically, it is a high-molecular-weight polymer of β-D-glucopyranose whose linkages are about 30% β-(1→3) and 70% β-(1→4). The (1→4)-linked β-D-glucopyranosyl residues (cellulose-like portions) occur in groups of three or four, separated by single (1→3) linkages [3], which make the molecule flexible and soluble, different from cellulose [4]. Due to this chemical feature, when ingested in foods, β-glucan forms highly viscous solutions [4] that can trap glucose and cholesterol and reduce their absorption in the gut [2,5]. Barley β-glucan is, therefore, soluble dietary fiber [5,6] with hypoglycemic and hypocholesterolemic effects [2,5,7,8].

In general, β-glucans can be classified according to their source as either cereal or non-cereal, which differ in their branching patterns, as the former are unbranched whereas the latter usually have 1→6 linked branches off the main side chain [6]. These structural differences confer different health benefit profiles, which vary from the anti-tumor and immunomodulatory effects described predominantly for fungal β-glucans, to metabolic effects such as improved glycemic control and cholesterol-lowering effects for cereal β-glucans [6]. Cereal β-glucans also perform different biological actions as prebiotic polysaccharides [9]. Thus, β-glucan is regarded as an important functional ingredient for the cereal foods industry, and the inclusion of β-glucan in both cereal and dairy-based food systems has gained interest [1,5]. This is mainly because, in developed countries, people tend to consume excessive amounts of carbohydrates, proteins and lipids, which increase their risk of suffering from chronic diseases (obesity, type 2 diabetes mellitus, and coronary heart disease [10]). Reducing cholesterol and post-prandial blood glucose levels have, therefore, become medical targets of relevance for public health [5].

The level of β-glucan in the grain is strongly dependent upon the genotype [1]. Hull-less barleys often have higher β-glucan contents than hulled feeding barley, and the former is mainly used as a human food because of its easier edibility [11]. Breeding for barley with high β-glucan content is, thus, an important task. Individuation of the progenies with the highest levels of β-glucan is hampered by the long time required for assaying β-glucan. Hence, the use of an indirect and fast determination procedure based on near-infrared reflectance spectroscopy (NIRS) is of great help in barley [12] as well as in oats [13].

NIR spectroscopy is a well-established method for rapid quality control in the food industry, to determine proximate chemical compositions (e.g., protein, dry matter, fat, and fiber) of a wide range of food ingredients and products [14]. NIR is a spectroscopic technique that measures the absorption of NIR radiation (from about 800 to 2500 nm) by the sample [15]. The wide applicability of NIRS in the agroindustry is due to the fact that the NIR region gives information about overtones and combination tones (stretching and bending) involving bonds with hydrogen (such as C-H, O-H, N-H, and S-H), characteristic of organic compounds [14,16,17].

For using NIRS, an appropriate calibration model has to be developed for each product (i.e., for a given material in a given physical state; for example, either intact grains or ground grains) to relate the compound of interest (parameter) to the sample’s spectral data [15,17]. To this aim, useful spectral information can be extracted by modern chemometrics methods, where “chemometrics” is defined as the application of complex mathematical and statistical methods to chemical data [15]. In general, NIRS provides more precise results when ground samples are used, because of their better homogeneity [15]. Even for β-glucan predictions, making NIRS assessments on the whole grain flour produces better results than on intact grains [18]. Interest in the assessment of barley β-glucan content with NIRS has produced stimulating results over the years, with increasing accuracy of measurements for ground barley grains [12,18,19,20,21,22,23]. Specifically, Albanell et al. [12] recently obtained good predictions of the β-glucan content (R^2^ = 0.912 for validation) in the 1.18%–9.95% range, thanks to the availability of advanced instrumentation. In the present work, we used the same NIRS device those authors found to be best for the indirect determination of the grain β-glucan content, albeit without the visible light. In fact, to determine β-glucan content, the NIR region (1108–2492 nm) provides the best accuracy of the model [12]. Furthermore, by using barley genetic materials with enhanced β-glucan content, we were able to extend the range of prediction to 3.4%–17.6%, since a wide range of analyte content across samples is recommended to develop NIRS predictive models [16]. Indeed, one major problem to construct a NIRS calibration for β-glucan content has been the limited β-glucan range for the calibrations [22].

## 2. Materials and Methods

### 2.1. Barley Samples

A set of 248 barley samples was used in this study. In order to represent the most common variations in grain types observed in commercial and breeding genotypes (see [1]), such set comprised genotypes with both morphological diversity for the ear type (i.e., both two and six-rowed barleys were included; where the latter have smaller kernel size, higher husk content and higher protein content than two-row barleys) and for grain type (that is, both hulled and hull-less genotypes were present; where the latter have a lower content of crude fiber since they lack the hull covering the caryopsis), as well as genotypes with either a normal starch (about 75% amylopectin and 25% amylose) or a *waxy* endosperm (that is, barleys whose starch has a very low, or null content of amylose). The sample set consisted of two groups of barley genetic materials: 185 breeding lines (the progeny of a single cross) with a very wide range of β-glucan contents (such cross produced transgressive lines with very high β-glucan contents: we are currently using this progeny for genetic mapping, and a manuscript with details about this genetic material is under preparation), and 63 samples of feeding and malting barley cultivars (with low to medium β-glucan contents). The former group ensured a high variability in β-glucan (necessary to develop a good predictive model), whereas the latter was introduced to make the calibration set more representative of the spectral diversity that can be encountered in barley. Although this latter device is not expected to improve the validity of β-glucan predictions, broadening the sample population to include the greatest possible spectral variability is recommended to improve the general applicability of the predictive model [14,16]. Breeding lines were grown in a greenhouse (92 in 2019 and 93 in 2020) at Fiorenzuola d’Arda (Northern Italy), whereas barley varieties were grown in fields either at Fiorenzuola d’Arda or were obtained from other Italian locations in 2020 and 2021. Barley grain samples were ground to 0.5 mm with a Cyclotec Sample Mill (Foss Italia S.p.A., Padova, Italy) for NIRS analysis. A portion of each sample flour was used for the laboratory reference analysis.

### 2.2. Chemical Analysis

The laboratory reference analysis for β-glucan content was carried out with a mixed-linkage β-glucan assay kit (K-BGLU; Megazyme, Bray, Ireland) according to the streamlined procedure of McCleary and Codd [24]. This method is one of the most widely used direct assays for the measurement of β-glucan content, and it has been adopted by AOAC International (Method 995.16), AACC (Method 32-23.01), and ICC (Method No. 166). Absorbance was measured at 510 nm with a DU-730 spectrophotometer (Beckman Coulter Inc., Brea, CA, USA). According to the manufacturer’s instructions, standard errors of ±3% of the β-glucan content are routinely achieved with this method. For this analysis, water content was determined with a moisture analyser (HA60 IR, Precisa Instruments, Diekinton, Germany). β-Glucan content was expressed as a percentage of the whole grain flour, on a dry weight basis (dwb).

### 2.3. NIRS Analysis

For NIRS analysis, spectral data were collected (in duplicate scans for each unique sample, a procedure that can improve the reliability of NIRS calibrations [17]) with a NIR DS2500 F spectrometer (FOSS, Hillerød, Denmark) equipped with a monochromator with an approximate spectral range of 850–2500 nm and a dual Si and PbS detector, in reflectance acquisition mode (i.e., the radiation reflected from the sample surface was measured). Samples of whole grain flour (about 3 g) were analyzed by using a FOSS mini-open ring cup cell (38 mm internal diameter). Reflectance values (R) were recorded every 0.5 nm from 857.5 to 1091.5 and from 1107.5 to 2491.5 nm (the two intervals correspond to the Si and PbS detectors, respectively), according to the default setting, and automatically transformed to log (1/R) by the instrument software (ISIscan Nova; FOSS, Hillerød, Denmark). A log (1/R) transformation is typically carried out because, in this way, the measured reflectance is converted into an absorbance, which is proportional to the concentration of the absorbing species, thereby usually assuring linearity of spectral response to the trait of interest [25]. For NIRS, log(1/R) is commonly referred to as ‘pseudo-absorbance’ because, when working in diffuse reflectance, the path length is unknown and varies with the optical properties of the sample [25]. Each spectrum was collected as the average of five spatial sub-samples, assessed during the automatic rotation of the cup, each obtained from 32 sub-scansions, as per default. In general, we kept the default settings unless a reason to do differently was evident. We assumed that, in this way, the risk that our predictive model was overfitted to the data (by cherry-picking the best combination of settings) was reduced, and that its generalizability and reproducibility were, therefore, better.

### 2.4. Calibration Procedure

To obtain a calibration model for β-glucan, chemometric analysis was performed using the FOSS Calibrator 3.3 software (FOSS, Hillerød, Denmark). Prior to calibration development, spectral data are pre-processed in order to reduce the spectral interferences and to emphasize the spectral effects related to the sample composition [15,16,17]. In this study, the spectral pre-treatments were as follows: for normalization, log (1/R) spectra were transformed using the standard normal variate (SNV, which normalizes each spectrum by subtracting the spectrum mean and dividing by the standard deviation) and detrend (an algorithm that removes the basal trend from each spectrum) mathematical treatments to correct the baseline and reduce scattering interference, chiefly due to the particle size of the flour [26]; then, as the differentiation method, the Norris-Williams algorithm was used [17,27]. For differentiation, either the default 1,16,16,1 or a 1,4,4,1 (as recommended by [12]) derivative transformation was applied for signal enhancement and correction of baseline drift and peak overlapping [15,17]. These mathematical derivatives are defined as follows [17]: the first number is the order of the derivative, the second is the gap over which the derivative is calculated, and the third and fourth digits are the number of data points used for the first and second smoothing, respectively. The first derivative is commonly used [17,27], and it was specifically shown to be an optimal spectral pre-treatment for β-glucan prediction [12]. As the derivative magnifies fine spectral undulations, it can amplify the noise present in the spectrum [25]; hence, a first smoothing is typically applied prior to the derivative calculation in order to decrease the detrimental effect on the signal-to-noise ratio that the derivative would, otherwise, cause [27]. A second smoothing can then be applied, but, in both derivative transformations, the last digit of 1 indicates no second smoothing was applied [17] in the present case. Application of the second derivative was tested too [18].

To detect the presence of outliers, Principal Component Analysis (PCA) was performed over the whole spectral data set prior to building any predictive model with modified Partial Least Squares (mPLS) regression [25]. In this respect, PCA represents a qualitative method of exploratory data analysis that transforms a set of correlated real variables (pseudo-absorbance values at each measured wavelength, in this case) into a new set of uncorrelated (orthogonal) variables (principal components), which are linear combinations of the original ones and are constructed to capture as much as possible of the variability present in the original variables [16]. Solely the variability present in the spectra is therefore considered in this analysis, independent of any measured parameter. Each successive principal component accounts for as much of the remaining spectral variability as possible so that the first ones contain most of the useful information. Hence, the complexity of the data can be reduced by retaining only the principal components that capture the largest part of the existing spectral variation. This small number of principal components can thus be used to extract information from the data much more easily than it could be done on the original, untransformed (and correlated) variables [25]. To this aim, the default spectral pre-treatments (namely, SNV, detrend, and the 1,16,16,1 Norris–Williams algorithm) were used, and 16 principal components were retained, as by default. The samples were evaluated for spectral outliers according to GH values, which correspond to standardized versions of Global Mahalanobis distances, where standardized Mahalanobis distances (H) are distances among sample spectra in the spectral data space of reduced complexity (i.e., with 16 principal components, in this instance). Specifically, GH values are standardized Mahalanobis distances between each sample spectrum and the center of the whole sample set (that is, the average sample spectrum) [15].

Afterward, the whole set of spectra used was automatically divided into two subsets, one for training and one for validation (comprising 80% and 20% of the unique spectra, respectively; as per default), by randomly assigning the samples to each set with the constraint that the same distribution of β-glucan contents across the selected samples was maintained in the two subsets [16]. In this way, 198 unique samples (each corresponding to a pair of spectra) of the whole set of 248 barley samples described in Section 2.1. (Barley samples) were used for training and 50 for validation (here intended as test set validation, that is, the predictive model is tested on another set of samples, different from those used to develop the model [28]). Spectra duplicates were automatically held together in the same set (i.e., paired) for training, cross-validation, and validation, to preserve the independence of subsets [16]. Prediction models were computed with mPLS regression, assuming a linear relationship between β-glucan content and its absorbance—or pseudo-absorbance—in the ground barley grain [25]. As in other chemometric methodologies, in mPLS regression, the spectral data are reduced to a few independent mathematically-constructed factors, holding the decisive spectral information [16,25]. The mPLS factors are latent variables construed as orthogonal (uncorrelated) linear combinations of the original variables (i.e., the spectral responses at each measured wavelength). Analogously to the principal components of the PCA, mPLS factors account for the variability existing in the multidimensional spectral dataspace, but, differently from the principal components of the PCA, they maximally capture the covariance between pseudo-absorbances and the reference values of the trait to be predicted [25]. A model complexity (or dimensionality) of 16 was eventually used; that is, the mPLS regression models were developed over the main 16 mPLS factors. This dimensional reduction—which, at present, matches the maximum number of mPLS factors established according to the default setting of the FOSS Calibrator software—represents a trade-off between precision of calibration and risk of overfitting [25], which both increases with the number of mPLS factors used. Although using a high (>10, indicatively) number of mPLS regression factors is a potential cause of overfitting [16,17], which reduces the model robustness for future predictions [27], recent technological advances have greatly improved the signal-to-noise ratio. This enables better exploitation of spectral nuances captured by minor mPLS factors (those that explain only a small portion of the spectral variance), with reduced concerns of overfitting the data to noise rather than to real features of the spectra. Predictive models with 15 or more mPLS factors ought, nevertheless, be carefully validated to avoid the risk of overfitting [17].

Cross-validation was performed with a Venetian-blinds method using four data blocks. FOSS Calibrator software provides performance statistics about developed predictive models: bias, slope, intercept, R^2^, sᵣ, SEP, and RMSEP (see [25] for a description). The bias is the difference between the average of actual (reference) values and the average of predicted values, and, therefore, it represents a measure of accuracy (how close a set of measures are to their true value). Slope and intercept refer to the linear regression of actual versus predicted values, and they should ideally be 1 and 0, respectively. They too, particularly the former, represent measures of accuracy. In the calibration process, optimization of the multidimensional mPLS regression constrains the predicted data to minimize their dispersion around the linear regression of actual versus predicted values, and therefore, the slope and intercept of the linear regression of actual versus predicted values give an indication of the goodness of the optimization of the multidimensional mPLS regression [16]. R^2^, the coefficient of determination, is a measure of fitting, which is the opposite of dispersion; that is, it expresses which fraction of the variation in the predicted data is captured by the model [25]. Hence, it chiefly expresses precision, that is, the closeness of predicted data to each other along with the calibration regression and, therefore, their closeness to the linear regression of actual versus predicted values, as calibration optimization minimizes the dispersion of the predicted data around the linear regression of actual versus predicted values [16]. Another useful parameter is sᵣ (aka SD_rep_), the standard deviation between duplicate NIRS measurements, pooled across samples; it represents the repeatability standard deviation (SD_rep_), that is, the precision of spectral assessment. RMSEP is the Root Mean Squared Error of Prediction and accounts for the total error, including all random and systematic errors, and thus represents an overall measure of accuracy (inclusive of precision) of the predictive model. The standard error of prediction (SEP) accounts for the random errors only, and thus it provides a measure of the precision of the predictive model. All these statistics are separately calculated for calibration, cross-validation, and validation.

For validation, observed deviations of the regression parameters, slope, and intercept, from their ideal values (1 and 0, respectively) were tested for significance of difference (at the *p* = 0.05 threshold) according to [29] (Equation (17.18)), after calculating their standard errors of estimate with equations 17.20 and 17.27 of [29], respectively. Significance of bias was tested (at the *p* = 0.05 threshold) with a paired-samples *t*-test [29] between measured (reference) versus predicted values.

## 3. Results and Discussion

Table 1 shows the compositional data of the ground barley samples employed for the development of the training and validation sets in this study.

Log (1/R) spectra for ground barley samples are shown in Figure 1. The obtained NIRS raw spectra are very similar to those previously reported for this product [12,20,21,30,31]. Basically, they are also quite similar to the typical spectrum of ground wheat (see [17] for a typical wheat NIRS spectrum). The characteristic bands for β-glucan, as studied in barley, are in the regions from 1194 nm to 1240 nm and from 2260 nm to 2380 nm [20,22,30].

The samples were then evaluated for the presence of spectral outliers based on GH values [15,17]: a GH value higher than three is commonly considered indicative of a potential outlier when a predictive model is used, and a GH value higher than five is used to identify potentially faulty samples (e.g., a wheat sample that has been erroneously included together with the barley samples) when a predictive model is developed. Very few samples exceeded the former but not the latter threshold (Figure 2) and were then physically inspected. As they were not found to be faulty, they were retained to build the β-glucan predictive model, since the inclusion of (unfaulty) samples with high GH values can improve the stability of the calibration [17]. In general, although erroneous samples (e.g., pear in a set of apples) have a serious negative impact on the resulting predictive model, extreme samples truly belonging to the target population can be highly informative in the model-building phase [25].

As expected, increasing the complexity of the model (i.e., the number of mPLS factors retained) increased the estimated accuracy and precision of the model, that is, RMSEP(cross) decreased (Figure 3). Indeed, in order to decide on the model complexity, it is necessary to quantify its effect on the model performance [25]. As the linear mPLS regression aims at minimizing the sum of squared errors between actual (reference) β-glucan contents and β-glucan content predictions based on the mPLS factors, a performance metric related to this prediction sum of squares, i.e., RMSEP, is the most obvious choice to evaluate the model performance [25] as it includes the systematic predictive error, that is, the bias [16]. To this aim, cross-validation is useful to find the optimal number of mPLS factors to be included in the predictive model—based on RMSEP(cross)—while avoiding overfitting. In fact, RMSEP(cal) always improves (decreases) as mPLS factors are added, but this is because more and more predictive factors can be used in the regression to predict smaller and smaller fluctuations in the data which, ultimately, leads to overfitting [16,17]. This eventually results in modelling the calibration onto noise because of an overabundance of mPLS factors, as the mPLS factors, explaining a smaller and smaller portion of spectral variability, end up accommodating spectral noise rather than bear some, albeit low, valuable spectral information. In other words, as the model complexity increases the apparent fitting of the model also increases whereas its predictive capability decreases [16]. Instead, RMSEP(cross) provides an overall estimate of the prediction error that is not over-optimistic like RMSEP(cal), and, like RMSEP(val), RMSEP(cross) typically passes through a minimum as the number of variables is increased [16]. Moreover, RMSEP(cross) estimation is averaged across blocks (four in this case), thereby reducing stochastic variations associated with smaller subsets of samples. This could affect RMSEP(val) instead, since a single subset is used for validation, and might prematurely stop the inclusion of further mPLS factors. Although the use of cross-validation to simulate validation has been criticized, and the use of an independent set of samples has been advocated to be a better way to decide on model complexity [25,28], chiefly because bias—and therefore accuracy—estimation is more reliable [16], even Esbensen and Geladi [28] conceded that, at least from a general viewpoint, cross-validation is a legitimate approach for model comparison. This is because it furnishes a uniform framework within which alternative parameter settings can be objectively compared for model performance. In this way (i.e., using RMSEP(cross) to decide the number of mPLS factors), even model complexity is first established on a sample set and, then, it is tested on another sample set, thereby fully corroborating the general validity of the developed predictive model. Indeed, the validation of a model must include validation of the complexity of that model, which is a key determinant of the model itself [16,17].

In the present case, RMSEP(cross) invariably decreased, and it was still decreasing when the 16th mPLS factor was introduced, thereby suggesting that all these mPLS factors added some spectral information useful to improve the predictive model [16]. Indeed, the FOSS calibrator software uses all the mPLS factors necessary to reach the lowest RMSEP(cross), within the maximum number established in the settings. The maximum number of mPLS factors was, however, not raised above the default setting (i.e., 16), to prevent overfitting, which could reduce the generalized applicability of the model to unobserved samples. Schmidt et al. [18] used a similar maximum number of mPLS factors (namely, 15), and all of them were included in their best-performing predictive models, obtained with a Fourier transform near-infrared reflection instrument. However, those authors did not carry out a validation of their predictive models, which is subsequently necessary to evaluate overfitting [17,28]. In this respect, it can be worth noticing that, for a given predictive model, a range of mPLS factor complexities is commonly observed, across which, for all practical purposes, alternative picks are equally good within existing uncertainty bounds [28].

As implemented by the calibration procedure, the distributions of β-glucan contents among samples were similar in the training and validation sets (Figure 4).

On the training set, calibration of the β-glucan predictive model was performed, and cross-validation was further applied to obtain a rough evaluation of the model repeatability, and, therefore, of its accuracy (closeness between predicted and actual data across the calibration regression, that is, trueness of predictions).

For calibration, the linear regression of actual (reference) versus predicted values is shown in Figure 5, together with the plot of residuals, which were normally distributed. The model fit (Table 2) was very good: R^2^ = 0.994. This is slightly better than the β-glucan predictive model found by Albanell et al. [12], which had an R^2^ = 0.966 for calibration. As expected for a good calibration, bias and regression intercept were both zero, and the regression slope was equal to one. The standard deviation between duplicate measurements sᵣ represents the precision of spectral assessment. Although it was small, 0.148% (dwb), it was not negligible with respect to both the standard error of prediction (SEP, 0.257%) and the root mean squared error of prediction (RMSEP, 0.257%). SEP is calculated as the standard deviation of the random differences (i.e., those that correspond to deviations from the regression line) between NIRS-predicted and reference values after correction for bias (that is, bias is removed, so that SEP is a pure measure of precision), whereas RMSEP includes all random and systematic (i.e., affecting the whole regression) errors, that is, it includes the bias [17,25]. For calibration, RMSEP is identical to SEP as no bias exists. Thus, as sᵣ was non-negligible with respect to SEP, including duplicate scans for each sample is confirmed to be useful to improve the precision of predictions. RMSEP, which is an estimate of the overall error of the predictive model, that is, of the variation in β-glucan content unexplained by the predictive model, represents an important statistic to be compared across calibration, cross-validation, and validation for the evaluation of the predictive model.

For cross-validation (Table 2), changes in the regression parameters were minimal: bias = 0.006%, slope = 1.006, intercept = −0.042% and R^2^ = 0.990. However, very little information on the reliability of slope and bias (and, therefore on the real accuracy of the predictive model) is provided by cross-validation [17,28]. The estimate of sᵣ was increased only slightly (from 0.148% to 0.154%), confirming that the variation in the spectral duplicate is small but non-negligible, and, therefore, measurement precision benefits from collecting two (or even more) spectra by physically resampling (ideally, without replacement, that is, collecting replicate spectra from different subsamples) each batch [17]. More interestingly, the loss of performance for SEP and RMSEP was small: they both increased to 0.329% (with a tiny difference for the fourth decimal figure, as bias was no longer zero). Precision—for which cross-validation provides information that is more reliable than the information it provides about accuracy—was, therefore, still good, indicating that calibration statistics were not over-optimistic. Therefore, the prediction model was not overfitted to the training set. This, however, was already apparent from Figure 3, where such information was used to evaluate the model complexity.

Although calibration and cross-validation give a reasonable account of the precision that can be reached with a predictive model, the estimated statistics based on the training set, particularly those expressing accuracy, tend to be optimistic since the predictive model was built-up, entirely (calibration) or partially (cross-validation), on these very same data. Hence, more reliable estimates of statistics describing the applicability of a new predictive model are provided by testing it on an independent set of samples, i.e., by validation. Thus, the linearity of response (slope and intercept), accuracy (bias), fitting (R^2^), precision (SEP), and overall fit (RMSEP) estimates for the validation set represent the real assessment of the model. Specifically, accuracy can only be appraised, in terms of bias, by validation [17,28].

For the validation set, the linear regression of actual (reference) versus predicted values, and the relative residuals plot, are displayed in Figure 6. Even in this case, residuals were normally distributed. Table 3 shows the performance statistics: applying the predictive model to the validation dataset increased the bias to only 0.013% (not statistically significant), which confirms the great accuracy of the β-glucan predictive model. Slope (1.006) and intercept (−0.039) deviated only minimally from ideal values (neither deviations were statistically significant), indicating a linear response of NIRS β-glucan predictions even over this extended range of β-glucan contents. R^2^ was the same as for cross-validation: 0.990. In general, the present study confirms and extends previous findings of NIRS prediction of β-glucan content in barley, such as those reported by Schmidt et al. [18] using a Fourier transform near-infrared reflection instrument (R^2^ = 0.952–0.989 for calibration), by Seefeldt et al. [22] with dispersive NIRS (R^2^ = 0.94 for cross-validation in the 1194–1240 nm region of the spectrum), and, of course, by Albanell et al. [12] who found an R^2^ = 0.912 for validation. As said, the better performance of our predictive model with respect to those studies, specifically that by Albanell et al. [12], is mainly due to the wider range of β-glucan contents in our samples (3.4%–17.6%), a range comparable to that used by Seefeldt et al. [22], who, however, did not carry out a validation of their predictive model and used only three genotypes at different grain filling stages. A broad range of the studied component is, of course, useful to improve the fitting of a NIRS model [16,17]. For illustrative purposes, the overall (training plus validation sets) linear regression of actual (reference) versus predicted values is shown in Figure 7.

As previously said, mPLS regression requires that the relation between the spectral variables and the trait of interest is linear. Whereas mPLS regression can cope with some mild nonlinearities by including more mPLS factors in the model, it cannot cope with strong nonlinearities unless model complexity becomes unacceptable [16]. Such large nonlinearities can be detected as deviations from a straight line in a plot of the reference values against the corresponding predicted values for the quality trait of interest [25], though such a plot allows only a rough evaluation of linearity [16]. Figure 7, therefore, demonstrates that, with the model complexity used, β-glucan content displays a sufficiently linear spectral response in the ground barley grain, as no large nonlinearities are evident.

As for the other statistics of validation performance (Table 3), the estimate of sᵣ increased to 0.197%, confirming the usefulness of spectral duplicates. Most importantly, SEP and RMSEP were even better than for the cross-validation set: 0.328% and 0.327%, respectively. They still had almost identical values because the prediction bias was minimal (and nonsignificant), whilst a decrease in the random error with respect to cross-validation can only be due to stochastic variations between the average characteristics of the training and validation sets since validation with a new test set always tends to result in a higher RMESP estimate than any cross-validation alternative [28]. This confirms that not only the accuracy but also the precision of the β-glucan predictive model was high (as already shown by cross-validation statistics), with no overfitting. Overall, this empirical match indicates that no relevant difference in the covariance data structure was present between the training set and the validation set [28], that is, β-glucan content was associated with the same spectral features in the two sample sets. All this supports great reliability for the predictive model developed in this work.

It is interesting to compare the RMSEP values with the standard error of the reference method: as the latter is of the order of ±3% of the β-glucan content (that is, it is roughly proportional to β-glucan content itself, and, therefore, it is heteroskedastic), and the average β-glucan content in the present study was 8.7% (Table 1), it can be calculated that the average standard error of the reference method should be around 0.261%. RMSEP(val) was 0.327% (Table 3), only slightly higher than the standard error of the reference method, with no evident heteroskedasticity (Figure 7). This indicates that on average, NIRS adds only a small error to the assessment of the β-glucan content above what is provided by the reference method. This additional error tends to be noticeable with respect to the reference method for low β-glucan contents, but much less so for high β-glucan contents (as the NIRS error is constant, whereas the error of the reference method increases with β-glucan content). This represents a useful condition for the evaluation of barleys with high β-glucan content, the genetic materials for which this predictive model assumes importance.

Furthermore, as Albanell et al. [12] suggested that a Norris–Williams 1,4,4,1 derivative transformation is an optimal spectral pre-treatment to develop a β-glucan predictive model, we compared such transformation with the default 1,16,16,1 transformation (which was not tested by [12]). The performance of validation for this additional predictive model is given in Table 4. The bias was slightly higher than that obtained with the default transformation, but the slope and intercept were very close. R^2^ was the same. RMSEP was minimally increased, reflecting the larger bias. Basically, the two transformations produced predictive models with almost coincident performances, with a tiny improvement for the 1,16,16,1 transformation, which was, therefore, preferred.

For a better comparison with the β-glucan predictive model developed by Albanell et al. [12], samples with a β-glucan content overlapping the range used by those authors were selected from our sample set (in practice, we used only the samples with ≤10% β-glucan) to build up a ‘reduced’ predictive model with the same spectral pre-treatments used for our own ‘full’ model. To keep the number of training samples close to that used for our ‘full’ model, all the 183 unique samples thereby selected (corresponding to the 3.4–10.0% range) were used and no validation was performed. For this model, 15 mPLS factors were automatically used (based on the RMSEP(cross)). This ‘reduced’ predictive model had RMSEP = 0.247, R^2^ = 0.98, slope = 1.01, intercept = −0.08, and bias = −0.003 for calibration. Hence, the value of RMSEP(cal) was slightly smaller than that found for the model based on the full sample set (possibly because of the greater precision of the reference method in this lower range), but the linear fitting (i.e., the R^2^) was also lower than for the ‘full’ model (see Table 2), which is quite obvious because a narrower range typically results in a smaller fraction of the variation existing in the predicted data that is captured by the model [25]. Both parameters evidenced better precision with respect to the predictive model of Albanell et al. [12] (who found RMSEP = 0.332 and R^2^ = 0.967 for calibration). Although a genuine comparison is not possible because the actual samples were different between the two studies (and the precision of the reference values could differ as well), this result may reasonably be ascribed to the higher number of mPLS factors used in our ‘reduced’ predictive model, i.e., 15, with respect to that used by Albanell et al. [12], that is, nine. The slope, intercept and bias were good, though not ideal as, instead, observed for the β-glucan predictive model developed on the full sample set. As expected, a higher number of mPLS factors tends to increase the model precision (see Figure 1).

Finally, as Schmidt et al. [18] preferred to use the second derivative for differentiation, we checked this data pretreatment too. Table 5 shows the performance of validation for this further predictive model. The bias was higher than that obtained with the default transformation (see Table 3), and the intercept was slightly farther from zero, but the slope was quite close. The R^2^ was the same, but the RMSEP was higher, reflecting the larger bias. Using either the first or the second derivative, therefore, did not cause large differences in the performances of the predictive models, but the first derivative was better. We also tested the whole 2,5,5,1 Norris–Williams algorithm as a differentiation method, since it is the one originally used by Schmidt et al. [18]. Its performance, however, was even worse than that obtained with the 2,16,16,1 differentiation method (e.g., RMSEP(val) = 0.417, R^2^ = 0.98), and, therefore, it was dismissed. Noteworthy, both these algorithms ended up using 15 out of the maximum of 16 mPLS factors, like the best predictive models originally developed by Schmidt et al. [18]. Eventually, the current default settings were confirmed to provide an optimal predictive model for barley β-glucan content.

With respect to the enzymatic laboratory reference analysis (the mixed-linkage β-glucan assay kit), our predictive model for β-glucan content has some interesting advantages (once the NIR spectrometer is available, which is the chief restraint to the use of this technology) that are resumed in Table 6, and correspond to the general reasons for using an indirect NIRS method of measurement [14,15,16,17].

In conclusion, this and other recent studies on the application of NIRS for measuring β-glucan content of ground barley grains demonstrate the great effectiveness of this indirect measurement technique to assess such grain components, much better than in the past. This is due to the high signal-to-noise ratio reached with the most recent instruments, which also allows using a higher number of mPLS factors—even those bearing a small load of the total explained variance—so to capture even minor nuances of the spectra and thus better exploit the information they contain without incurring in overfitting. Ultimately, however, though developing a predictive model for β-glucan content is very useful for dedicated breeding programs, the availability of high-β-glucan genetic materials is a prerequisite for the development of a predictive model suitable for the indirect measurement of high β-glucan contents. Progress, indeed, occurs by the concomitant advancement of multiple conducive conditions.

## Figures and Tables

**Figure 1 foods-11-01846-f001:**
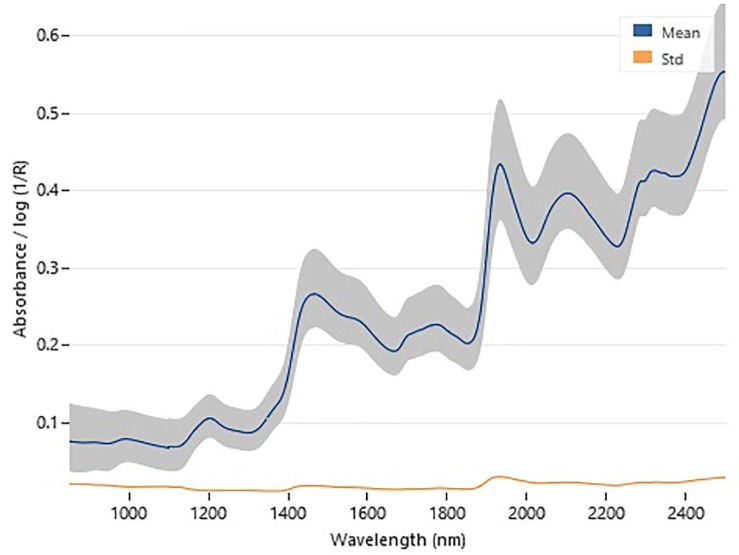
Average, range and standard deviation of the NIRS spectra (857–2492 nm) of the samples.

**Figure 2 foods-11-01846-f002:**
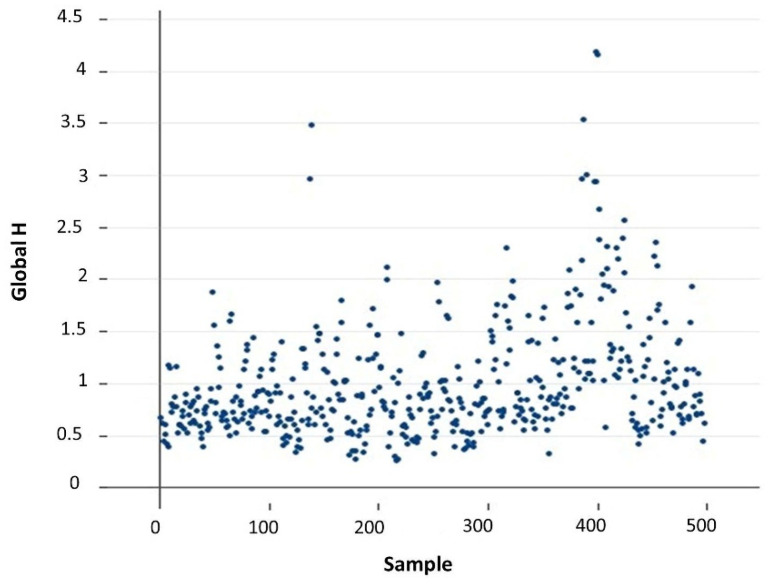
Global Mahalanobis (GH) distances of samples’ spectra from the average spectrum.

**Figure 3 foods-11-01846-f003:**
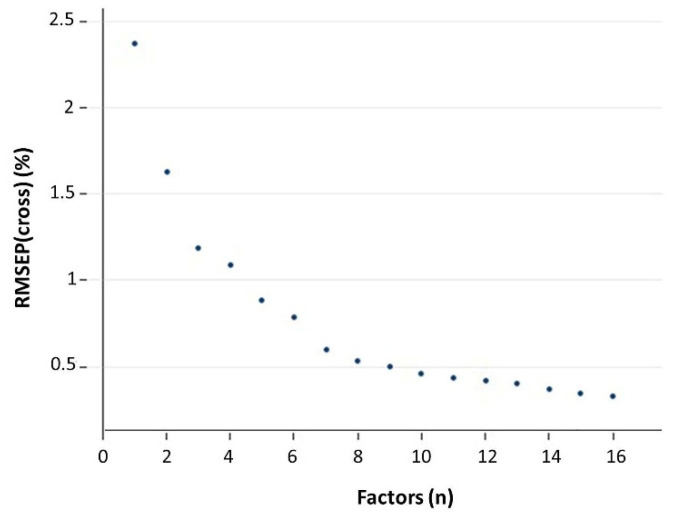
Root mean squared error of prediction (RMSEP) for cross-validation versus the number of mPLS factors used for regression.

**Figure 4 foods-11-01846-f004:**
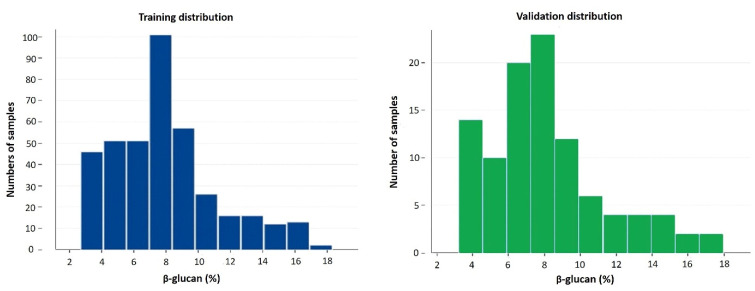
Histogram distributions of β-glucan contents (g/100 g, dwb) in the training (**left**) and validation (**right**) sets of duplicate samples (i.e., counts consider that two paired spectra were collected for each unique sample).

**Figure 5 foods-11-01846-f005:**
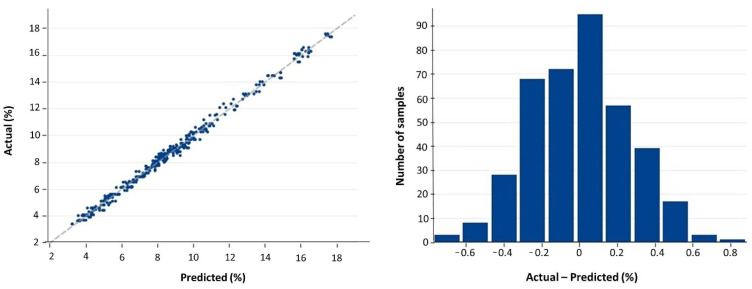
Plot of measured (actual) versus predicted β-glucan contents (g/100 g, dwb) for the training (calibration) set (**left**), with its frequency histogram plot of residuals (**right**). Datapoints and frequencies refer to the duplicate scans (i.e., the two paired spectra that were collected for each unique sample).

**Figure 6 foods-11-01846-f006:**
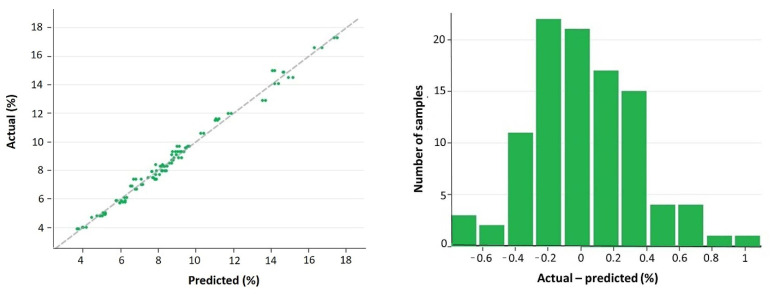
Plot of measured (actual) versus predicted β-glucan contents (g/100 g, dwb) for the validation set (**left**), with its frequency histogram plot of residuals (**right**). Datapoints and frequencies refer to the duplicate scans (i.e., the two paired spectra that were collected for each unique sample).

**Figure 7 foods-11-01846-f007:**
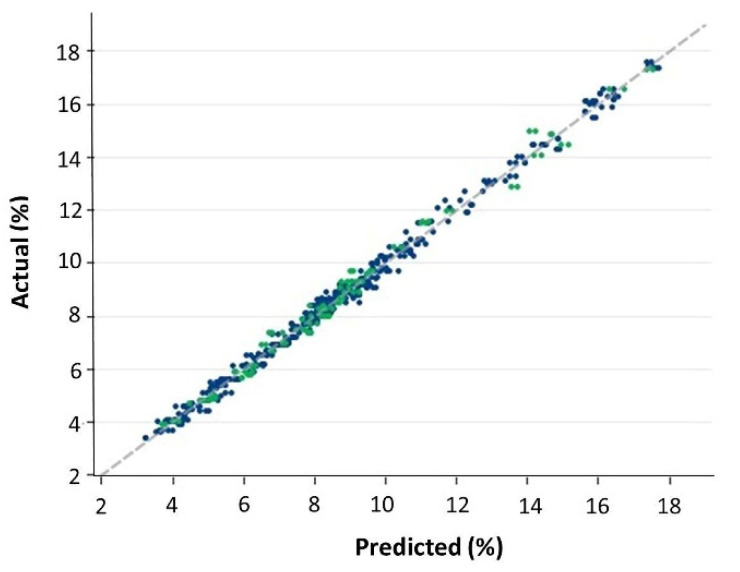
Linear regression plot of measured versus predicted values of the whole (training plus validation) dataset for β-glucan content (g/100 g, dwb). Blue dots correspond to the training set and green dots to the validation set. Datapoints refer to the duplicate scans (i.e., the two paired spectra that were collected for each unique sample).

**Table 1 foods-11-01846-t001:** The number of samples used to develop the β-glucan predictive model, and basic statistics about the β-glucan content (g/100 g, dwb) of the samples used.

Set	Unique Samples	Minimum	Maximum	Mean
Training	198	3.4%	17.6%	8.7%
Validation	50	3.9%	17.3%	8.7%
Overall	248	3.4%	17.6%	8.7%

**Table 2 foods-11-01846-t002:** Training performance.

	Bias	Slope	Intercept	R^2^	sᵣ	SEP	RMSEP
Calibration	0.000	1.000	0.000	0.994	0.148	0.257	0.257
Cross-validation	0.006	1.006	−0.042	0.990	0.154	0.3291	0.3287

**Table 3 foods-11-01846-t003:** Validation performance.

Bias	Slope	Intercept	R^2^	sᵣ	SEP	RMSEP
0.013	1.006	−0.039	0.990	0.197	0.328	0.327

**Table 4 foods-11-01846-t004:** Validation performance of the model with 1,4,4,1 differentiation method.

Bias	Slope	Intercept	R^2^	RMSEP
0.019	1.006	−0.03	0.99	0.328

**Table 5 foods-11-01846-t005:** Validation performance of the model with 2,16,16,1 differentiation method.

Bias	Slope	Intercept	R^2^	RMSEP
0.03	1.01	−0.06	0.99	0.381

**Table 6 foods-11-01846-t006:** Comparison between NIRS and reference enzymatic methods for some characteristics relevant to their routine applicability.

NIRS	Comparison Criteria	Enzymatic Assay
+++	Instrumentation costs	+
+	Analytical cost per sample	+++
+++	Speed	+
+++	Throughput level	+
+	Expertise level	++
+	Work-flow complexity	++

## Data Availability

Not applicable.

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
