# Peer review of "Indirect Measurement of β-Glucan Content in Barley Grain with Near-Infrared Reflectance Spectroscopy"

_foods, 2022, doi:10.3390/foods11131846_

Round 1
Reviewer 1 Report
This work is interesting for it can save time for measuring the β-glucan content. However, as I know, the β-glucan content of barley grain normally ranges from 2-9% around. You have to make an explanation about your samples. Do you have a series of dilutions that show the same fold increase or decrease by your method? You are also missing the direct testing with other similar methods in order to illustrate the advatages or improvements of your established method.
Author Response
Reviewer’s comment: This work is interesting for it can save time for measuring the β-glucan content.
Authors’ response: Thank you very much for appreciating our work!
Reviewer’s comment: However, as I know, the β-glucan content of barley grain normally ranges from 2-9% around. You have to make an explanation about your samples.
Authors’ response: although a few mutants with high β-glucan content are available, the reviewer is fully right in saying that “the β-glucan content of barley grain normally ranges from 2-9%”. Nevertheless, our Centre has performed barley breeding for more than forty years, and a breeding program for high β-glucan content was undertaken too. Thus, we found a cross that produced transgressive lines with very high β-glucan contents. We are currently using these materials for genetic mapping, and a manuscript will, hopefully, be produced in the next few months. This point has been clarified in the manuscript: “… 185 breeding lines (the progeny of a single cross) with a very wide range of β-glucan contents (such cross produced transgressive lines with very high β-glucan contents: we are currently using this progeny for genetic mapping, and a manuscript with details about this genetic material is under preparation)”.
Reviewer’s comment: Do you have a series of dilutions that show the same fold increase or decrease by your method?
Authors’ response: Serial dilutions are very important to demonstrate linearity of response when an analyte solution is used to build up a measurement curve, that is, a concentrate solution of the analyte to be measured is progressively diluted with pure solvent to produce an array of concentrations. Although this can also be done for a NIRS predictive model by mixing a high β-glucan standard, or a high β-glucan genotype flour, with a low-content one in various proportions, developing a predictive model based on many independent samples is typically preferred for NIRS (see [15,16]). This is because grains are a complex matrix wherein spectral interference effects are predominant over the analyte spectral effect and, therefore, to be usable for real samples a predictive model must be developed on samples really representative of the product that will be analysed (see [15,16,25]). In this respect, linearity of response (see Figures 5-7) is better demonstrated by using real samples rather than mixtures of the analyte, which do not account for interferences of the actual complex matrix, or of the variability inherent to it, with the features of the spectrum. To this aim, several levels of analyte content must be available, otherwise, linearity cannot be properly judged by fitting to a regression line throughout the whole range. We believe that our materials provided an array of β-glucan contents that is good to judge the fitting of predicted vs measured data to a regression line throughout the whole range. This was, indeed, the most innovative aspect of our work.
Reviewer’s comment: You are also missing the direct testing with other similar methods in order to illustrate the advatages or improvements of your established method.
Authors’ response: We are a bit puzzled by this remark: the NIRS predictive model has been developed on the basis of reference measures of β-glucan content made with a direct method. This was described in section ‘2.2. Chemical analysis’: “The laboratory reference analysis for β-glucan content was carried out with a mixed-linkage β-glucan assay kit (K-BGLU; Megazyme, Bray, Ireland) according to the streamlined procedure of McCleary and Codd [24]”. This method is one of the most widely used direct assays for the measurement of β-glucan content, and, as specified in the manufacturer’s instructions, it has been successfully evaluated by AOAC International (Method 995.16), AACC (Method 32-23.01) and ICC (Method No. 166). We have now mentioned these details in section 2.2. We have also added, in the Discussion section, a new Table to illustrate the advantages of NIRS method over this direct assay.
Reviewer 2 Report
This study focus on the application of NIRS for measuring β-glucan content of ground barley grains which demonstrate a progressive effectiveness.
The idea of this article is relatively novel, and it proposes a good indirect determination of the grain β-glucan content. This approach is very interesting.
However, this article still has some deficiencies as follows:
line 91, on the sentence "both two and six-rowed, hulled and hull-less, with normal starch and waxy," It seems that the explanation is not clear enough.
line 191, A set of 248 barley samples were conducted in current experiment with 185 breeding lines , and 63 samples of feeding and malting barley cultivars, then,the authors mentioned that 198 unique samples were used for training and 50 for validation.Is there a correlation between the two sets of numbers?
Figure 4, histogram distributions showed duplicate samples, Should have also applied double sample size in Figure 5 & 6, however, it did not mentioned.
Author Response
Reviewer’s comment: This study focus on the application of NIRS for measuring β-glucan content of ground barley grains which demonstrate a progressive effectiveness.
The idea of this article is relatively novel, and it proposes a good indirect determination of the grain β-glucan content. This approach is very interesting.
Authors’ response: We wish to thank the Reviewer for acknowledging the interest of our approach.
Reviewer’s comment: However, this article still has some deficiencies as follows:
line 91, on the sentence "both two and six-rowed, hulled and hull-less, with normal starch and waxy," It seems that the explanation is not clear enough.
Authors’ response: As mentioned in the manuscript, “broadening the sample population to include the greatest possible spectral variability is recommended to improve the general applicability of the predictive model [14,16]” (lines 99-100). Barley displays major morphological diversity for several traits, one main division is in two general types: six-row barley and two-row barley. They differ for the arrangement of the spikelets/grains alongside the ear rachis: the former type has six rows of kernels along its length, whereas the latter has only two rows on opposite sides of the ear, because, in these barleys, the other spikelets are small and sterile and, therefore, they do not produce grains. Two-row barley has larger kernel size, lower husk content and lower protein content than six-row barley. Besides, though most barley cultivars have covered/hulled grains, i.e., the outer hull remains attached to the kernel after harvesting, some cultivars are hulless (or naked), that is, the hull is not adherent to the kernel and is thus removed during the harvest. This causes a noticeable difference in the grain composition. Moreover, starch represents the chief component of the barley kernel, and it is, in turn, comprised of amylose and amylopectin, which differ for chain length and branching. Mutations at the waxy locus are among the main mutations affecting the barley kernel: waxy genotypes have a starch without amylose (i.e., made of amylopectin only). Furthermore, as mentioned on line 94, “feeding and malting barley cultivars”, which have different physiological and compositional features, were included too. The first three traits can be present in every possible combination, but hulless barleys are most often used for human nutrition; six-row covered barleys are used for feeding, whereas two-row covered barleys with normal starch (non-waxy) are commonly used for malting purposes. Together, these four grain traits represent a large portion of the most common variations observed in commercial and breeding genotypes. A useful predictive model could be developed separately for each of them, but as several combinations of these traits are possible, it would be ideal if a single predictive model can be developed for them all. This was, indeed, our purpose. We have better explained these aspects in the manuscript.
Reviewer’s comment: line 191, A set of 248 barley samples were conducted in current experiment with 185 breeding lines , and 63 samples of feeding and malting barley cultivars, then,the authors mentioned that 198 unique samples were used for training and 50 for validation.Is there a correlation between the two sets of numbers?
Authors’ response: Yes, the 248 barley samples mentioned on line 91 (section ‘2.1. Barley samples’) and the 198+50=248 unique samples described on lines 191-192 (section ‘2.4. Calibration procedure’) are the same. We have highlighted this relationship by better specifying it in section ‘2.4. Calibration procedure’ as follows: “In this way, 198 unique samples (each corresponding to a pair of spectra) of the whole set of 248 barley samples described in section 2.1. (Barley samples) were used for training and 50 for validation …”. We hope that this amendment can solve any misunderstanding or lack of clarity.
Reviewer’s comment: Figure 4, histogram distributions showed duplicate samples, Should have also applied double sample size in Figure 5 & 6, however, it did not mentioned.
Authors’ response: Many thanks for pointing out this imprecision! We have now clarified this aspect in the legends of Figures 5-7 by adding the following sentence: “Datapoints and frequencies refer to the duplicate scans (i.e., the two paired spectra that were collected for each unique sample)”. This is because deviations between measured and predicted values are better evidenced without spectral averaging since mistakes can occur during the collection of each individual spectrum.
Round 2
Reviewer 1 Report
Reviewer’s comment: You are also missing the direct testing with other similar methods in order to illustrate the advatages or improvements of your established method.
Authors’ response: We are a bit puzzled by this remark: the NIRS predictive model has been developed on the basis of reference measures of β-glucan content made with a direct method. This was described in section ‘2.2. Chemical analysis’: “The laboratory reference analysis for β-glucan content was carried out with a mixed-linkage β-glucan assay kit (K-BGLU; Megazyme, Bray, Ireland) according to the streamlined procedure of McCleary and Codd [24]”. This method is one of the most widely used direct assays for the measurement of β-glucan content, and, as specified in the manufacturer’s instructions, it has been successfully evaluated by AOAC International (Method 995.16), AACC (Method 32-23.01) and ICC (Method No. 166). We have now mentioned these details in section 2.2. We have also added, in the Discussion section, a new Table to illustrate the advantages of NIRS method over this direct assay.
I mean the comparison with the similar methods, such as Albanell et al., (2021), especially the samples within the same range of both methods.
Author Response
[PREVIOUS] Reviewer’s comment: You are also missing the direct testing with other similar methods in order to illustrate the advatages or improvements of your established method.
[PREVIOUS] Authors’ response: We are a bit puzzled by this remark: the NIRS predictive model has been developed on the basis of reference measures of β-glucan content made with a direct method. This was described in section ‘2.2. Chemical analysis’: “The laboratory reference analysis for β-glucan content was carried out with a mixed-linkage β-glucan assay kit (K-BGLU; Megazyme, Bray, Ireland) according to the streamlined procedure of McCleary and Codd [24]”. This method is one of the most widely used direct assays for the measurement of β-glucan content, and, as specified in the manufacturer’s instructions, it has been successfully evaluated by AOAC International (Method 995.16), AACC (Method 32-23.01) and ICC (Method No. 166). We have now mentioned these details in section 2.2. We have also added, in the Discussion section, a new Table to illustrate the advantages of NIRS method over this direct assay.
Reviewer’s comment: I mean the comparison with the similar methods, such as Albanell et al., (2021), especially the samples within the same range of both methods.
Authors’ response: we have added a paragraph (lines 493-515) where we provide a more direct comparison with the predictive model of Albanell et al. (2021) by using a ‘reduced’ predictive model based only on “the samples within the same range of both methods”. Although a genuine comparison is not possible because the actual samples were different between the two studies (and the precision of the reference values could differ as well), we noticed a better precision of our ‘reduced’ predictive model with respect to the predictive model of Albanell et al. (2021). This result may reasonably be ascribed to the higher number of mPLS factors used in our ‘reduced’ predictive model. This is fully consistent with our conclusions that “Progress, indeed, occurs by the concomitant advancement of multiple conducive conditions” and, specifically, improvements conveyed by our predictive model were due to both “using a higher number of mPLS factors” as well as to “availability of high-β-glucan genetic materials”.